# A2BCF: An Automated ABC-Based Feature Selection Algorithm for Classification Models in an Education Application

Leila Zahedi [1,2,*], Farid Ghareh Mohammadi [3] and Mohammad Hadi Amini [1,2,*]

1 Knight Foundation School of Computing and Information Sciences, Florida International University, Miami, FL 33199, USA
2 Sustainability, Optimization, and Learning for InterDependent Networks (Solid) Laboratory, Florida International University, Miami, FL 33199, USA
3 Department of Computer Science, University of Georgia, Athens, GA 30602, USA; farid.ghm@uga.edu
* Correspondence: lzahe001@fiu.edu (L.Z.); moamini@fiu.edu (M.H.A.); Tel.: +1-(786)-828-8302 (L.Z.); +1-(305)-348-9936 (M.H.A.)

**Abstract:** Feature selection is an essential step of preprocessing in Machine Learning (ML) algorithms that can significantly impact the performance of ML models. It is considered one of the most crucial phases of automated ML (AutoML). Feature selection aims to find the optimal subset of features and remove the noninformative features from the dataset. Feature selection also reduces the computational time and makes the data more understandable to the learning model. There are various heuristic search strategies to address combinatorial optimization challenges. This paper develops an Automated Artificial Bee Colony-based algorithm for Feature Selection (A2BCF) to solve a classification problem. The application domain evaluating our proposed algorithm is education science, which solves a binary classification problem, namely, undergraduate student success. The modifications made to the original Artificial Bee Colony algorithm make the algorithm a well-performed approach.

**Keywords:** AutoML; Artificial Bee Colony; classification; education; evolutionary computation; feature selection; optimization; swarm intelligence; wrapper method; student success





## 1. Introduction

Student retention is a major concern for STEM fields. It is particularly problematic in computing fields, where enrollment has not kept pace with industry demands. Therefore, finding patterns in historical educational data can help the education community reveal the potential reasons for students' withdrawal from the computing field. This information provides guidelines to better understand the relative success of computing students and enable strategic solutions to achieve higher retention rates [1].

The application of ML in education is currently very interesting to researchers and education communities. Building predictive models involves learning from data. Machine Learning (ML) is a technology that enables computers to learn without being explicitly programmed. One of the most important recent discussions in this field is related to the enhancement of ML algorithms' performance in different applications, such as education. In the context of ML, choosing the best model has always been a concern. Moreover, it is crucial to transfer the most important information to both save time and improve performance. In general, based on the principle of Occam's razor, "a model should be simple enough for efficient computation and complex enough to be able to capture data specifics" [2].

The extraction of useful information and the presentation of scientific, educational decision-making is necessary, providing professionals with an additional source of knowledge. The educational datasets can be massive. The high-dimensional nature of many modeling tasks has given rise to a wealth of feature selection techniques.

Feature selection is a way of removing noise and random errors in the underlying data. In feature selection, we use techniques to select features that are more relevant to the problem and withdraw redundant or irrelevant data without incurring much loss of information [3]. Feature selection facilitates the best performance for the ML model [4]. The main idea of feature selection is to select a subset of features for the model to either improve the model's performance or reduce the structure's size and computational cost [5]. In [6], Koller and Sahami presented a filter feature selection method to reduce the overall computation cost. In Ahmed et al. [7], authors concluded that information entropy can determine the importance of features and can be effective for the reduction of attributes. Choosing K out of N features helps the training time to be distributed by N/K [8]. In feature selection problems, the size of the search space for a dataset with $N$ features would be $2^N$ [9]. Therefore, researchers have provided different search methods to reduce models' computational costs. Dash and Liu believed these methods typically have four basic steps: (1) a generation procedure to generate the next candidate, (2) an evaluation function, (3) stopping criteria, and (4) a validation procedure [10]. Cleaning and featurization of the data are considered the most time-consuming steps of AutoML, and researchers have investigated different methods to improve the performance of the AutoML process [11].

There are various heuristic search strategies to address combinatorial optimization challenges. These algorithms include, but are not limited to, nature-inspired algorithms such as Genetic Algorithms (GA) and Swarm Intelligence (SI), which include Particle Swarm Optimization (PSO) [12] and Artificial Bee Colony (ABC) [13].

This paper proposes a new approach (A2BCF) for feature selection using an ABC algorithm to improve the performance of the classifier in an education application and predict students' success. The goal of A2BCF is to choose a subset of features from the whole feature space, reducing the total number of used features to improve the total computation time. It is specifically helpful for ML algorithms, such as support vector machines, as their training time is highly dependent on the dimensions of the data.

Our main contributions are listed as follows:

- A2BCF is a novel framework for efficiently selecting the features in a large structured dataset.
- The application of the developed optimization method is used for the first time in this paper.
- We design a variant of the ABC algorithm that helps balance the exploration and exploitation phases of the ABC algorithm.

The remainder of the paper is organized as follows: Section 2 covers a summary review of feature selection. Section 3 presents the original ABC. Section 4 introduces ABC-based feature selection, our modifications, and the proposed algorithm. Following that, Sections 6 and 6.4 discuss the experimental methodology and results. Finally, Section 8 concludes this work.

## 2. Related Work

Recently, there has been an increase in the number of research studies that use dimensionality reduction as a preprocessing step to separate the noise and unimportant data from the important data. These methods include both feature selection and feature extraction methods. Feature extraction methods are the techniques used to extract new features from the data and are mostly used in image processing techniques. However, feature selection techniques are approaches that select an optimal subset of features from existing features with minimum error and information loss [14]. Unlike feature extraction techniques, feature selection methods are leveraged to structured datasets that have identified features. Therefore, this paper focuses on feature selection methods.

*Feature Selection*

Feature selection means finding a subset of features from the feature search space. Different techniques for feature selection include sequential forward selection (SFS) or

sequential backward selection (SBS). The former begins from the minimum number of features and adds features during the selection process, while the latter starts from the whole feature set and discards the irrelevant feature during the selection process [15,16]. At the end of the process, the remaining subset of features in both methods is considered the optimal subset. Recently, there has been an increase in robust approaches based on evolutionary algorithms that decrease the number of features. In [17], Faraoun and Rabhi used GA for dimensionality reduction, improving the accuracy of the classification process. In another study, Aghdam et al. [18] proposed an Ant Colony Optimization (ACO) approach for the feature selection process. This study shows that ACO demonstrates lower computational complexity than stochastic algorithms and GA.

On the other hand, there are two major categories of feature selection, Filter and Wrapper methods [19]:

- *Filter methods:* In these methods, selecting the features is performed before leveraging the learning algorithm. Therefore, the features do not depend on the ML algorithm. In these methods, the importance of features is calculated (based on some predefined criterion), and then the best feature subset is selected. In these methods, no learning algorithm is used for choosing the features. The advantage of these methods is their faster speed.
- *Wrapper methods:* Unlike filter methods, wrapper methods generate different subsets of features by adding and removing features to achieve reasonable accuracy. The predictive accuracy of the classifier is used to evaluate the subset of features. The advantage of these methods include their high classification accuracy. However, their computational complexity is higher than filter methods. Therefore, researchers have been exploring different methods to enhance the convergence of wrapper methods. Wrapper methods have gained much attention due to their promising performance. Prior works have used wrapper methods for selecting the optimal features [8,20–23].

There has also been an increase in the use of evolutionary algorithms, specifically SIs, for the feature selection process in the past few years [8,24–26]. Another wrapper feature selection method was proposed by Zawbaa et al. and Ng et al. using Bat Optimization algorithms [25,26]. ABC algorithm is also an SI algorithm that has been used for feature selection problems [8].

## 3. Artificial Bee Colony Algorithm

ABC, initially introduced by Karaboga [27], is one of the most recent SI methods that simulate the foraging behavior of honey bees. In ABC, the bee colony has three groups of bees: (1) employed, (2) onlookers, and (3) scouts. **Employed bees** have the responsibility of exploiting the food sources and sharing the information they gather with **onlooker bees**. Onlooker bees may or may not select a food source based on the received information. The higher the quality of a food source, the more it is likely to be chosen by the onlookers. Onlookers are also responsible for exploitation. On the other hand, **Scout bees** control the exploration process. If the scout bee finds a quality food source, it becomes an employed bee. In contrast, if a food source becomes thoroughly exhausted, the involved bee becomes a scout. Although Scout bees have low investigation cost, the mean of the food sources they find is low [8]. Algorithm 1 shows the main steps of the ABC search strategy inspired by [8].

---

**Algorithm 1** Search process by Artificial Bee Colony

---

1: Initialize the population (size = PS)
2: Evaluate the population
3: **while** Stopping criterion is not met **do**
4:    Assign each food source to an employed bee for exploitation in the neighborhood
5:    Onlooker bees choose food sources based on the information shared
6:    Scout bees search the area randomly to find quality food sources
7:    Memorize the best food source
8: **end while**
9: **return** Return best food source

---

As can be seen in the algorithm, the first steps include initialization and evaluation of the population. Next, employed bees start exploitation in the neighborhood using Equation (1).

$$V_{i,j} = X_{i,j} + rand(-1,1)(X_{i,j} - X_{k,j}) \tag{1}$$

where $k$ is an index for one of the features and $rand(-1,1)$ is a random number uniformly distributed in the range $[-1,1]$. Then, the better food source between $V_{i,j}$ and $X_{i,j}$ is chosen using greedy selection. Afterwards, employed bees share the gathered information with onlooker bees. Onlooker bees start calculating the probability values using roulette wheel selection, as shown in Equation (2).

$$P_i = \frac{f_i}{\sum_{j=1}^{PS}(f_j)} \tag{2}$$

where

$$fit_i = \begin{cases} \frac{1}{1+f_i}, & f_i \geq 0 \\ 1 + abs(f_i), & f_i < 0 \end{cases} \tag{3}$$

Equation (3) shows how the fitness ($fit_i$) is calculated using the objective function ($f_i$). Since the performance of the classifier is always positive, we always have the first function in the equation. Therefore, in this problem, we are maximizing the objective function or, in other words, we are minimizing the fitness. If onlooker bees select a food source, Equation (1) and greedy selection are used again to generate and choose new food sources.

In the ABC algorithm, the scout bee is chosen from employed bees. This selection is made based on a control parameter called **limit**. Therefore, if a specific food source is not enhanced until a fixed number of iterations (limits), that food source is abandoned (or exhausted) by its employed bee and the employed bee turns into a scout. The abandoned food source is then replaced with a randomly generated food source using Equation (4) that may lead to discovering rich, unknown food sources.

$$X_{i,j} = x_{min} + rand(0,1)(x_{max} - x_{min}) \tag{4}$$

In the end, the location of the best food source is memorized and the algorithm starts from the beginning. This process continues until the stopping criterion is met. The stopping criteria could be a threshold for different metrics such as desirable performance, maximum number of evaluations/iterations, or time.

## 4. Feature Selection Using ABC

As mentioned in the previous section, in a feature selection problem using ABC, each solution (candidate food sources) is a vector with size $N$, where $N$ equals the number of features in the dataset. This vector is a bit vector that only consists of 0 and 1. If the value of a specific position (feature) is 1, the feature is to be included in the evaluation. On the other hand, if the corresponding value is 0, the feature would be excluded from the assessments.

Mohammadi et al. [28] motivated researchers to apply nature-inspired algorithms and evolutionary algorithms to solve large-scale optimization problems, specifically feature selection processes. In feature selection problems using ABC, the quality of each food source equals the accuracy of the classifier over the validation set with the corresponding subset of features.

Since the original ABC is only applicable to continuous problems, proper strategies should be adopted to transform it into a binary problem. The major steps to implement the ABC-based feature selection are as follows:

1.  **Create initial population:** First, a population of food sources is generated. Each food source in the population is a bit vector consisting of 0s and 1s.
2.  **Calculate fitness:** Each solution is submitted to the ML classifier and the accuracy is calculated. The accuracy is then saved as the fitness of the solution.
3.  **Exploitation by employed bees:** An employed bee takes each bit vector to regulate the neighbor food sources (bit vector). SFS or SBS are the common approaches used by an employed bee in this step to find better solutions [29].
4.  **Exploitation by onlooker bees:** The information about the performance of solutions is shared with onlookers, and they select the subsets with a better probability of exploration. Then, the chosen bit vector goes under the same process as step 3 to become exploited.
5.  **Memorizing the best food source:** After all the onlookers are done with their parts, the subset with the best quality is memorized.
6.  **Exploration by scout bees:** In this step, a scout bee is assigned to deserted food sources, if any, and a bit vector of size $N$ is generated randomly. This vector is then submitted to the ML classifier and its accuracy is stored.

The general structure and foremost steps of ABC-based feature selection are given in Figure 1.

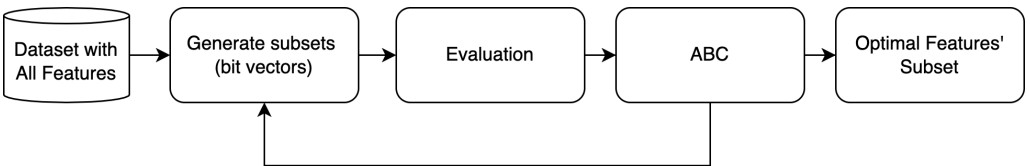

**Figure 1.** The general schematic approach to ABC-based feature selection.

To improve the efficiency of the original ABC approach, we made some further modifications to the algorithm. The main goal of this work is to provide optimized feature selection with a better accuracy rate. This work obtains the optimized feature subset by using Automated Artificial Bee Colony Feature Selection (A2BCF). The A2BCF algorithm selects the optimized features, and the efficiency is calculated using different ML classifiers.

One disadvantage of SI algorithms, including the ABC, is their premature convergence. These algorithms would be more efficient if their convergence rate were faster. The other challenge is the dependency of the algorithm on the initial population. In other words, if the proper initial population is selected, the algorithm reaches the optimal solution in a more reasonable time. There is an increased number of research studies exploring these challenges in order to enhance the convergence rate of these algorithms [30]. Balancing between the exploration and exploitation phases of such algorithms increases the efficiency of the algorithm. Therefore, ABC still has space for improvements. Karaboga and Akay [31] compared different variants of ABC algorithms and determined that ABC has relatively poor performance in the exploration phase. Hence, we consider improving the ABC algorithm's initial population and scout bee phase to design a more efficient ABC variant.

## 5. Proposed Feature Selection Method (A2BCF)

A2BCF algorithm is a novel ABC algorithm for feature selection, inspired by the OPT-ABC algorithm proposed in [32] for hyperparameter tuning purposes. The proposed

algorithm returns the ideal set of features that increases the classifier's accuracy. Figure 2 shows the schematic flow chart of A2BCF. In this algorithm, the selected classifier is leveraged on every bit vector of features (features' subset). Since the accuracy depends on the learning algorithms, the method is wrapper-based.

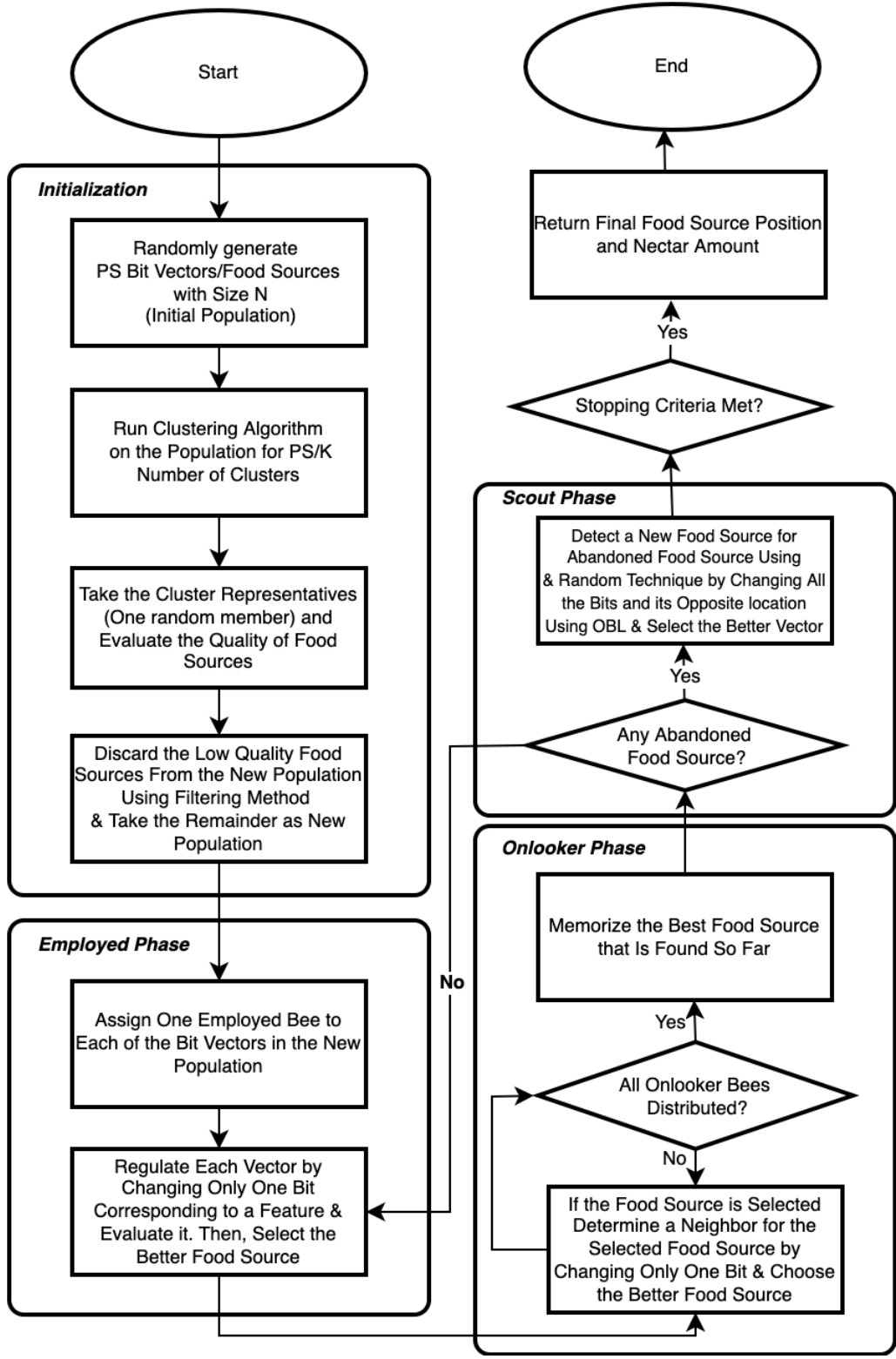

**Figure 2.** The flowchart of the proposed A2BCF algorithm (stopping criterion could be set to the maximum of number of evaluations or iterations).

The steps of the A2BCF are explained as follows:

- **Initialization:** Only some solutions are evaluated in this step instead of all the random initial population members. Hence, an agglomerative clustering technique using cosine similarity function is first employed to group the similar or repetitive members of the initial population into the same cluster. As shown in Algorithm 2, we create a population of random food sources (bit vectors). Subsequently, only the representatives (one random member) of each cluster are taken as our new population to be evaluated by the objective function. In other words, the new population is a diversified sample of bit vectors from the original population. The clustering method (agglomerative clustering) used in this study is the most common type of hierarchical clustering and, in our case, the cosine similarity function is used to group the similar objects. Therefore, the most similar bit vectors belong to the same cluster. This approach is inspired by Chavent [33] for grouping the binary data samples.

    After taking all clusters' representatives, inferior food sources are discarded (based on a predefined threshold) from the new population. This is performed by filtering the low-quality representatives to avoid further evaluation of poor food sources. A threshold of 50% is considered in this study to filter the low-quality food sources and focus on the food sources that have the potential to improve in a lower number of iterations. As explained in the feature selection steps using ABC, the algorithm changes only one bit of the vector to exploit the search space. Hence, if a bit vector has poor quality, it takes longer to exploit the food source and reach a high-quality food source. Therefore, early stopping of those food sources helps improve the convergence rate of the exploitation phase.

    Classification accuracy is computed using 3-fold cross-validation to improve the classifier's reliability. In k-fold cross-validation, the dataset is divided into k equally sized folds and the ML algorithm is executed three times. Cross-validation is a preferred method over a single train–test split that trains on multiple train–test splits and provides a better insight into how well a model performs on unseen data. Equation (5) below shows that we aim to minimize the classification error rate. The accuracy is used in the fitness function to evaluate the quality of food sources.

$$fit_i = \frac{1}{1 + f_i} \tag{5}$$

and

$$f_i = Accuracy = \frac{TP + TN}{TP + TN + FP + FN} \tag{6}$$

where TP, TN, FP, and FN stand for true positives, true negatives, false positives, and false negatives, respectively. The reason for selecting accuracy as the performance metric is that the Multiple-Institution Database for Investigating Engineering Longitudinal Development (MIDFIELD) dataset used in this study is balanced and has an almost equal number of observations for both classes (54:46 ratio). In binary classification problems, a balanced dataset is one where positive values are approximately same as negative values (in our case, the number of students who graduated/dropped out from a computing field). However, in cases dealing with imbalanced data, other measures such as F1 score are more appropriate and can be used instead of accuracy.

- **Employed bee phase:** Each bit vector is taken by an employed bee, where it regulates the food sources (bit vector) by flipping only one of the bit values in the vector. If the neighboring (new) vector has a better fitness, it gets replaced with the previous vector, and its corresponding accuracy gets stored. In other words, the employed bee evaluates the model's accuracy by including (bit 1) or excluding (bit 0) only one of the features and passing the required information to onlookers.

- **Onlooker bee phase:** The information about the quality of the food sources is shared with the onlookers, and they then select the food sources with a better probability of exploration (based on Equation (7)). Then, selected food sources go under the same

process as step 3 to become exploited. After all the onlookers are done with their parts, the food source with the best quality is memorized.

$$P_i = \frac{Fit_i - min(Fit)}{max(Fit) - min(Fit)} \tag{7}$$

- **Scout bee phase:** This phase has an additional step in A2BCF. In original ABC, a scout bee (assigned to the deserted food source, if any) generates a random novel food source of size $N$. While onlooker/employed bees change only one bit of the bit vector, scout bees change all the bits in the vector. As we mentioned earlier, scout bees have a low mean in the food source quality that they find. Therefore, we added another step in which Opposition-Based Learning (OBL) is submitted to the abandoned food source to generate an opposite food source location based on Equation (8). Then, the algorithm moves forward with better food sources from random and OBL methods. The generated bit vector is then submitted to the ML classifier and its accuracy is stored.

$$\tilde{x}_{i,j} = 1 - x_{i,j} \tag{8}$$

- **Termination process:** The employed, onlooker, and scout bee phases will continue until the algorithm reaches the defined maximum number of evaluations/runs. In this study we set the maximum number of evaluations to 5000.

---

**Algorithm 2** Agglomerative Clustering Algorithm for Initialization Phase

---

**Input:** Dataset, $K = 100$, Population size (PS)
**Output:** *Population* (Cluster centroid/representative)

1: Randomly generate a population of size *PS*, with food sources of size length(*Features_set*)
2: # Clustering the random population to k clusters
3: Compute the proximity matrix (cosine similarity matrix)
4: Let each vector be a cluster
5: **while** The remaining clusters are more than PS/k **do**
6:     Merge the two closest clusters
7:     Update the matrix
8: **end while**
9: Take cluster representatives (one random member of each cluster), where k is defined as the new *population*
10: **return** *population*

---

Algorithm 3 presents the pseudocode of the A2BCF algorithm explained above.

---

**Algorithm 3** Proposed Algorithm—A2BCF

---

    **Input:** *threshold* = 50, *limit* = 3, $max_{eval}$ = 5000, $trial_i$ = 0    **Output:** Optimal features subset

1: Call Algorithm 2 (PS = 2000; 5000; 10,000)
2: Discard the food sources with poor quality($<$ *threshold* $=$ 50) from new population
3: **while** Stopping criteria is not met(number of evaluations $<max_{eval}$) **do**
4:     **for** *i* to *population_size* **do**
5:         Employed bee regulates the bit vector and find a new vector *Ni* in the neighborhood. This is achieved by changing only one bit in the vector
6:         Train the ML model with the novel vector (subset of features selected)
7:         **if** the new vector has a better quality **then**
8:             Replace the new vector with the original vector
9:         **else**
10:             $trial_i$ +=1
11:         **end if**
12:     **end for**
13:     **for** *j* to *population_size* **do**
14:         Onlooker bee calculates the exploration probability (*Pi*) according to Equation (2)
15:         **if** $rand(0,1) > Pi$ **then**
16:             Onlooker bee regulates the current vector *Ni* by changing only one of the bits in the vector, which gives a new vector in the neighborhood *Mi*
17:             Train the ML model with the updated features subset
18:             **if** the new vector has a better quality **then**
19:                 Replace the new vector with the original vector
20:             **else**
21:                 $trial_i$ +=1
22:             **end if**
23:         **else**
24:             Onlooker disregards the features subset and moves to the next bit vector
25:         **end if**
26:     **end for**
27:     Memorize and update the best subset so far
28:     **if** $trail_i > limit$ (food source is exhausted) **then**
29:         Scout bee generates a "Random" bit vector based on Equation (4)
30:         Scout bee generates the "OBL" bit vector based on Equation (8)
31:         Select the subset that gives better fitness between Random and OBL vectors
32:     **end if**
33: **end while**
34: **return** Optimal subset of features

---

## 6. Experimental Setup and Results

The detailed steps of the A2BCF algorithm were given in the previous section. This section presents the study's methodology; the performance of our proposed algorithm, A2BCF; and the comparison of the results with a previous study [1]. A2BCF is a wrapper feature selection approach that improves the classification accuracy and enhances the search to keep relevant features in a reasonable time. The dataset used for this experiment contains around 45,000 observations, and the number of classes is two. The overall structure of the whole experiment is presented in Figure 3. The A2BCF algorithm proposed in this study is inspired by a Hyperparameter Optimization (HPO) method proposed in 2021 [32].

As mentioned above, we leveraged several ML classifiers to select the most important features of the dataset. The details of the classifiers and the feature selection steps are described in the section below.

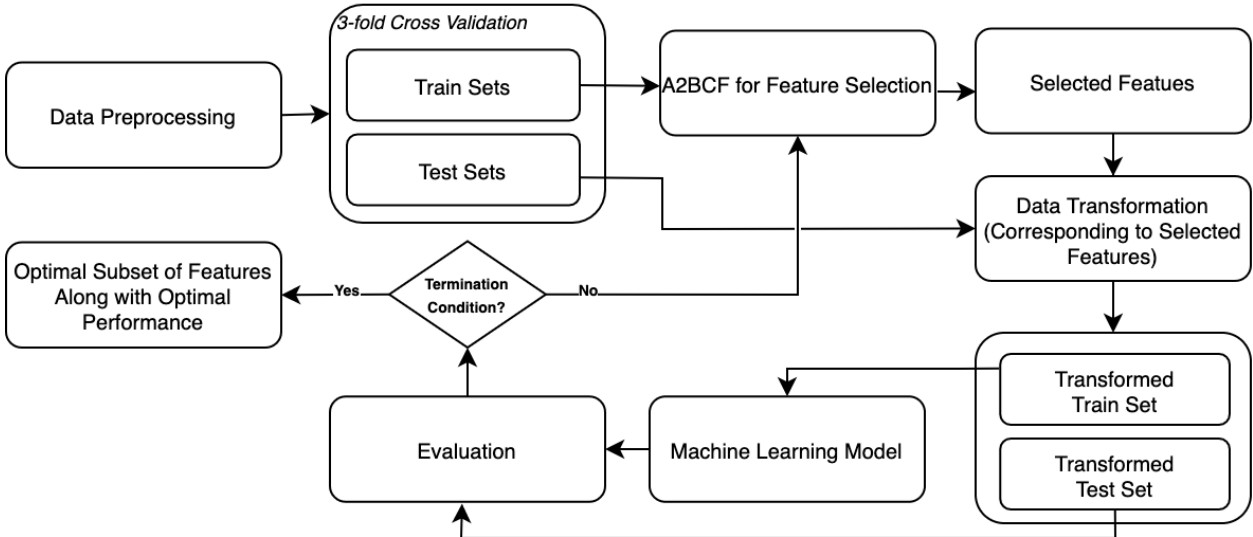

**Figure 3.** The overall structure of feature selection in this study.

### 6.1. Main Steps

The overall process of the study is developed in several steps, including data pre-processing, Feature Selection Optimization (FSO), data transformation, and training the transformed data. The details of these steps are explained below.

- **Data preprocessing:** This step is required to transform the data into an understandable state for the ML classifier. In other words, the raw data are manipulated into a form that can be accurately processed. Tasks such as data cleaning, feature engineering, and feature scaling are included in this step and summarized as follows:

  - Data cleaning: Data cleaning involves detecting and correcting/removing inaccurate records from the dataset. In this study, we detected the features with more than 60% missing values and removed them from the dataset. Moreover, the duplicated data were also removed from the dataset.

  - Feature engineering: In this study, this refers to creating new features from raw data to possibly increase the predictive power of the learning algorithm. Features such as students' entry term are examples of one of the created features. Additionally, one-hot encoding of the variables (for encoding the categorical types to their binary representations) is another feature engineering process performed in this study.

  - Feature scaling: Also known as data normalization, feature scaling is the approach used to standardize the range of features of data. As the range of values for different features may vary widely, it is a necessary step in data preprocessing to prevent the algorithm from becoming biased towards the features with higher values in magnitude.

  - In addition to the above steps, filtering is another step we performed in the preprocessing phase. Since our target population are the students in computing [1], we filtered the dataset to only include students from computing majors (CP = 11) to predict their success in their programs.

- **Feature selection optimization using A2BCF:** Feature selection is a primary task in Automated ML that helps the model achieve two main goals: better performance and reduced computational time. This step is the study's main contribution, an iterative ABC-based process to find the optimal set of features in a dataset. The proposed algorithm (A2BCF) is an improved version of the Opt-ABC algorithm [32] used for optimizing the hyperparameter tuning of the ML algorithms [34,35]. The output in this step is a subset of features selected by the algorithm.

- **Data transformation:** This process converts data to the required format of our destination system. As shown in Figure 3, after preprocessing the original dataset, it is divided into splits of training and test sets (using k-fold cross-validation). Then, the A2BCF optimization process starts. Depending on the selected subset of features by the algorithm, original train and test sets are transformed to new train and test sets (corresponding to the chosen features). Once the transformed training and test sets are ready, we leverage the ML algorithm on the train set. Then, the results are evaluated and stored using the testing set. This process continues until the stopping condition is met. Finally, the highest average accuracy (over three folds) and the optimal set of features are reported.
- **Train the transformed data:** In this step, the desired ML algorithm is submitted, and the ML algorithm gets trained. Finally, the accuracy of the ML algorithm and final features are returned.

### 6.2. Machine Learning Classifiers

A summary of the ML classifier used in this experiment is as follows:

1.  Random Forest (RF) [36] is an ensemble method and a type of Decision Tree (DT) learner that operates by constructing many DTs in the training phase. This is the reason it is called 'forest'. The term 'random' is also used because the trees are built differently with random samples and random features to add diversity to the models and decrease the chance of overfitting [36–38]. Random forest initially uses the bagging method to combine the predictions from each tree and calculate the overall predictions.

2.  Extreme Gradient Boosting (XGBoost) [39] is a developed version of Gradient Boosting that utilizes a gradient boosting framework as an ensemble. One of the focuses of XGBoost is the efficiency and speed of the model that supports parallelization. XGBoost also attempts to prevent overfitting using Ridge and Lasso regularization. XGBoost trains the model iteratively, correcting or fixing the newer models in each iteration [39].

3.  Support Vector Machine (SVM) is a supervised learning technique that generates input–output mapping functions. The mapping function for this study is a classification function in which nonlinear kernel functions are used to transform input to a high-dimensional feature space. In this feature space, the data become more separable when compared with the raw input. SVM works by finding the maximum-margin hyperplanes between positive and negative observations. Using mapping function, SVM transforms the nonseparable feature to linearly separable features [40,41].

### 6.3. Setting

In order to evaluate the performance of the A2BCF algorithm, the MIDFIELD dataset is used [42], and the goal is to predict students' success in computing majors (CP = 11). Students' attrition in computing fields is particularly problematic in computing. As mentioned in the previous sections, the ratio of the positive and negative classes in this dataset is 54:46. In other words, the percentage of negative values (attrition rate) is 46 percent and the percentage of positive values is 54 percent, which is in line with students' withdrawal from computing fields. Therefore, finding patterns in historical data can help education researchers and computing committees to detect the students who are more likely to drop out of their programs. MIDFIELD is a unit-record longitudinal database for bachelors' students from 20 universities across the United States. This version of the MIDFIELD dataset is used for a binary classification problem and has 4532 samples with 91 features after one-hot encoding the features. This dataset includes students' GPA (Grade Point Average), CIP (Classification of Instructional Programs code, standing for the students' major during the term, which is expressed as IPEDS (Integrated Postsecondary Education Data System), and COOP (Co-Operative education program) indicating student's participation in a partnership between their academic institution and an employer to obtain practical

experience through rotations of course study and employment [43]. Further, the SAT (Scholarship Aptitude Test) and the ACT (American College Testing) are standardized tests used for college admissions in the United States. Additionally, demographic information of the students, such as age, gender, race, and citizenship, is also included in this dataset. Information such as institution, transfer status, and the terms registered in the program are also among the dataset features. In summary, MIDFIELD consists of all the students' information in their universities profiles.

All the data are preprocessed using the techniques explained in the previous section. The dataset features are taken as the input for the A2BCF algorithm. To evaluate the fitness function, we use three classifiers, and ABC parameters are set to Maximum-iterations = 100; Initial Population Size (PS) = 2000, 5000, and 10,000; k = 100, with an average run of 10 times. The experiment is performed using cross-validation (CV) to yield more robust results. CV (3-fold) splits the input data into training data and test data independent of each other. Although cross-validation may increase the computational time, it reduces the chances of overfitting and provides a more reliable model. In this study, we used parallel cross-validation, and the final accuracy would be an average of the accuracy for each of the folds. We used *Scikit-learn* libraries, along with other Python packages, to leverage the ML models. The study experiments were conducted using Python 3.8.5 on High-Performance Computational (HPC) resources.

### 6.4. Results

This section covers the final results using the proposed algorithm. The validating process of the suggested A2BCF algorithm was executed on MIDFIELD dataset [42,44]. We compare the performance of A2BCF with a previous study using the same number of folds as in CV (3-fold) [34] while using the MIDFIELD dataset to answer the same classification question.

In summary, the proposed algorithm removes irrelevant features from the dataset in an effort to improve the accuracy. Reducing the features in this step can help the training process, training time, and/or other iterative processes such as HPO as it reduces the structure's size. In addition, diversifying the population improves the algorithm's convergence rate.

The overall numerical results with the experimental setting of Section 6.3 are given in Table 1. According to the Table, in all cases, the classifiers' accuracy after applying the A2BCF algorithm improved. As mentioned earlier, the accuracy is tested under different populations for three different classifiers (RF, XGBoost, and SVM). The advantage of A2BCF is notable when the PS is relatively small (PS = 50). Figure 4 shows the impact of the PS on the accuracy of the three ML algorithms. As can be seen, the accuracy of classifiers in all three cases are at their highest when the PS equals 50, and it again drops when the PS increases. A potential reason is that the algorithm's number of iterations is lower when the stopping criteria (maximum number of evaluations) are met, and the algorithms do not have the chance to improve themselves so many times. As can be seen, setting the PS around 50 gives the algorithm the chance to improve its performance.

**Table 1.** Comparison of A2BCF under different population sizes with a previous study.

| | Accuracy (%) | | | |
|---|---|---|---|---|
| Classifier | Reduced Feature Set by A2BCF | | | Original Dataset [34] |
| | N = 20 | N = 50 | N = 100 | N/A |
| RF | 88.47 | **88.51** | 88.44 | 85.30 |
| XGBoost | 88.67 | **88.76** | 88.64 | 85.16 |
| SVM | 87.84 | **88.00** | 87.81 | 85.06 |

N is the secondary population size after clustering and 3-fold cross-validation.

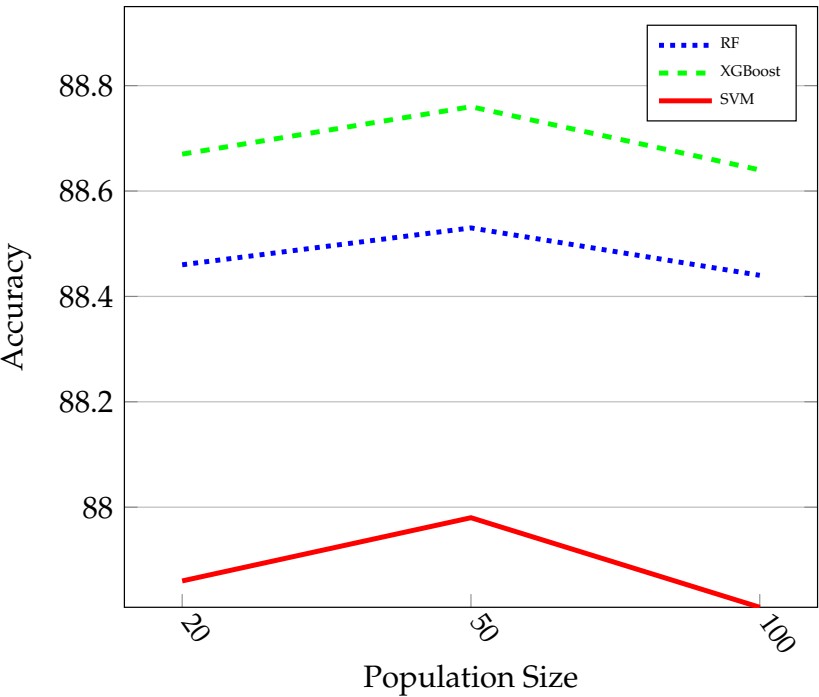

**Figure 4.** Impact of population size on classifier accuracy.

Among the three ML classifiers leveraged to predict student success, XGBoost achieves the best accuracy (88.76%) and is the most well-performed model. Hence, XGBoost is a good candidate for the final FSO framework in this application.

In summary, compared with a previous study [34] that considered the features without the FSO method, A2BCF outperforms in predicting student success. A2BCF uses an ABC-based approach to optimize data features and improve classification accuracy.

We also compared the results with another FSO method. Table 2 provides the comparison of the result between the PSO [45] and the proposed A2BCF method. The results show that A2BCF performs better than PSO.

**Table 2.** Comparison of A2BCF and PSO feature selection approaches.

| FSO Method | Accuracy (%) | | |
|---|---|---|---|
| | N = 20 | N = 50 | N = 100 |
| A2BCF | **88.51** | **88.76** | **88.64** |
| PSO [45] | 87.88 | 87.70 | 88.00 |

N is the number of food sources/particles in ABC and PSO approaches, respectively.

Based on the obtained results, it can be concluded that by leveraging A2BCF, we achieve better accuracy.

## 7. Discussion

In this work, an improved version of the ABC algorithm, A2BCF, is proposed for an automated feature selection approach in different ML algorithms. The proposed method is explored through a binary classification problem and tested on an educational application to predict undergraduate student success in 20 universities across the United States. The proposed method improves the exploration capability of the basic ABC algorithm by incorporating OBL learning algorithms. Further, the clustering algorithm used in the initial phase improves the population diversity and decreases the chance of searching potential areas. The obtained results are compared with those of a recent study, and the results show the robustness of A2BCF. The behavior of A2BCF is explored in different

conditions, and experimental results show that the algorithm improves the accuracy and can be employed to solve educational problems with relatively high dimensionality.

The main advantage of the proposed algorithm is its capability to find the optimal features in a reasonable time compared to exhaustive methods (such as grid search). A2BCF could be used as an ideal tool for preprocessing that optimizes the feature selection process, as it enhances the classification accuracy and minimizes the computational resources. Additionally, in this study, we are targeting to solve a specific problem in an education application. To this end, we focus on developing a tailored FSO algorithm that improves classification accuracy to predict student success. Based on the findings, it can be concluded that A2BCF is a promising approach for FSO to improve classification accuracy.

This study is a part of an ongoing research study to develop a robust AutoML framework that works well in different applications. Some future works are planned with the following directions. First, since we are only building classification models, investigating how our method can be extended to deal with regression problems can be helpful. Another suggestion is to evaluate the algorithm's efficacy on other datasets, which opens the opportunity to explore how different applications affect the performance of the A2BCF. Moreover, additional effort is needed to further improve the tuning time of such methods, including the A2BCF, without decreasing the achieved performance. Last but not least, we plan to insert the implementation of these modifications into our hyperparameter optimization frameworks [32,34,35] to further improve ML algorithms' performance in this study.

As mentioned earlier, the MIDFIELD dataset used is considered for predicting computing students' attrition rates in the computing field. Although the MIDFIELD dataset had been collected to denote the graduation rate of students and not for learning and teaching styles, our proposed approach is an automated and intelligent approach that can partly help education communities to provide guidelines for teaching and learning strategies.

## 8. Conclusions

Determining the optimum features is a crucial task in AutoML. In this work, the high-dimensional vector of 91 features from the combination of different features are reduced to fewer configurations using A2BCF and achieving the 88% classification accuracy. The A2BCF algorithm with different conventional ML classifiers is applied to the feature set to obtain the optimal or close to the optimal set of features. The results show that A2BCF yields better performance, with 88.76% classification accuracy compared with the accuracy over the original dataset and the PSO method.

**Author Contributions:** L.Z.: Conceptualization, methodology, software, validation, formal analysis, investigation, data curation, writing—original draft preparation. F.G.M.: Conceptualization, review & editing, M.H.A.: Conceptualization, resources, review & editing, methodology, supervision, project administration. All authors have read and agreed to the published version of the manuscript.

**Funding:** This research is supported through FIU's Dissertation Year Fellowships (DYF) funding.

**Institutional Review Board Statement:** Not Applicable.

**Informed Consent Statement:** Not Applicable.

**Data Availability Statement:** Not Applicable.

**Acknowledgments:** The authors acknowledge the University Graduate School (UGS) at Florida International University for providing financial support for this research in the form of Dissertation Year Fellowships (DYF) for Leila Zahedi. The authors would also like to gratefully acknowledge Matthew Ohland from Purdue University for providing access to the MIDFIELD dataset and the anonymous reviewers for the constructive comments for further improving the content of this article.

**Conflicts of Interest:** The authors declare no conflict of interest. The funders had no role in the design of the study; in the collection, analyses, or interpretation of data; in the writing of the manuscript, or in the decision to publish the results.

## Abbreviations

The following abbreviations are used in this manuscript:

| | |
|---|---|
| A2BCF | Automated Artificial Bee Colony for Feature Selection |
| ABC | Artificial Bee Colony |
| ACO | Ant Colony Optimization |
| CV | Cross Validation |
| FSO | Feature Selection Optimization |
| GA | Genetic Algorithm |
| HPC | High-Performance Computational |
| HPO | Hyperparameter Optimization |
| ML | Machine Learning |
| RF | Random Forest |
| MIDFIELD | Multiple-Institution Database for Investigating Engineering Longitudinal Development |
| OBL | Opposition-Based Learning |
| PS | Population Size |
| PSO | Particle Swarm Optimization |
| SBS | Sequential Backward Selection |
| SFS | Sequential Forward Selection |
| SI | Swarm Intelligence |
| SVM | Support Vector Machine |
| XGBoost | Extreme Gradient Boosting |

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
