# Peer review of "A2BCF: An Automated ABC-Based Feature Selection Algorithm for Classification Models in an Education Application"

_applsci, doi:10.3390/app12073553_

Round 1

Reviewer 1 Report

Point 1: In the manuscript titled “A2BCF: an ABC-based Feature Selection Algorithm for AutoML” the authors developed a feature selection algorithm with an automated artificial bee colony-based algorithm, called A2BCF, and evaluated the algorithm on education science. I think that this work was carefully conducted. The methods used are appropriate, and the data interpretation is convincing.

Point 2: The A2BCF could be provided for applications.

Reviewer 2 Report

The article has several issues preventing its publication in its current form. All of them are highlighted in the scanned version of the review.

Minor issues:

  • Use of English, grammar, formatting and ununified mathematical notation
  • Sentences without the proper references (in the scanned version appear under [ref])
  •  

Big issues:

  • Poor explanation of the algorithms and pseudocodes
  • Missing important explanations in the text, for instance: random number, following which distribution? Are you minimizing or maximizing? Accuracy computation over which set? and so on…
  • The biggest problem in my opinion is the experimental setup. As in the scanned version I consider using the full set (training and testing) to evolve a population and to perform feature selection, and then using the selected features to train and test the same classifier you use for evolution, and compare the same classifier before and after selection, is cheating. The proper setup should be dividing the dataset into training and testing, use the training set to obtain features (by evolution, or whatever procedure you want), and then compare the results before and after feature selection in the test set, that is, the feature selection procedure cannot see the test set ever.
  • Using just one dataset (for application purposes) if the aim of the paper is not to solve a specific problem but introducing a novel feature selection algorithm.
  • Having no comparison with respect to other algorithms for feature selection.

Again, the full set of questions are in the scanned version. I did not copy here because there are too many, I just prefer to upload the paper with the comments at margin.

Reviewer 3 Report

According to Author guidelines rewrite again.

Authors have used ABC algorithm as a feature selection tool , this can be compared with another algorithm such as PSO..

The paper is having some typographical mistakes

Discussion can be added with any similar recent works has been done on this.

Results are explained but explanation, can be elaborated in explanation if 3 methods used

Introduction can be elaborated

Round 2

Reviewer 2 Report

The authors tried to improve the paper since previous version, however, there are still some important issues unaddressed.

Section 4.

  1. First paragraph. Accuracy with respect to what? The training set (resubstituting error?), a validation set? Please clarify.

Section 5.

  1. Step initialization. How do you compute a centroid over binary data?
  2. The most similar. Using which similarity function?
  3. Algorithm 2. Same concerns as previous version
  4. Figure 2. Mostly the same concerns as previous. I am not able to implement this flowchart, due to it is not well explained. I the reader cannot implement, then it will be impossible to replicate. Please clarify.
  5. Equation 6. Please elaborate about data balancing

Section 6

  1. Figure 3 is better than previous, but still unclear. How about termination criterion? Please, provide a better explanation of the experimental setup.
  2. Algorithm 3. Line 2. How do you compute the threshold of a solution? Line 5. Which bit vector?
  3. Table 2. PSO? Which PSO? The binary one? There are many PSOs in the literature.

Section 7

  1. Execution time. You can conclude about a tradeoff involving execution time due to you did not present the time comparison analysis.

In addition, there are typos and other minor English errors. Please carefully correct them.
